# DNA binding polarity, dimerization, and ATPase ring remodeling in the CMG helicase of the eukaryotic replisome

Alessandro Costa[1]*, Ludovic Renault[1,2], Paolo Swuec[1], Tatjana Petojevic[3], James J Pesavento[3], Ivar Ilves[4], Kirsty MacLellan-Gibson[2], Roland A Fleck[2†], Michael R Botchan[3]*, James M Berger[5]*

[1]London Research Institute, Cancer Research UK, London, United Kingdom; [2]Department of Imaging, National Institute for Biological Standards and Control, Potters Bar, United Kingdom; [3]Department of Molecular and Cell Biology, University of California, Berkeley, Berkeley, United States; [4]Institute of Technology, Faculty of Science and Technology, University of Tartu, Tartu, Estonia; [5]Department of Biophysics and Biophysical Chemistry, Johns Hopkins University School of Medicine, Baltimore, United States

**Abstract** The Cdc45/Mcm2-7/GINS (CMG) helicase separates DNA strands during replication in eukaryotes. How the CMG is assembled and engages DNA substrates remains unclear. Using electron microscopy, we have determined the structure of the CMG in the presence of ATPγS and a DNA duplex bearing a 3′ single-stranded tail. The structure shows that the MCM subunits of the CMG bind preferentially to single-stranded DNA, establishes the polarity by which DNA enters into the Mcm2-7 pore, and explains how Cdc45 helps prevent DNA from dissociating from the helicase. The Mcm2-7 subcomplex forms a cracked-ring, right-handed spiral when DNA and nucleotide are bound, revealing unexpected congruencies between the CMG and both bacterial DnaB helicases and the AAA+ motor of the eukaryotic proteasome. The existence of a subpopulation of dimeric CMGs establishes the subunit register of Mcm2-7 double hexamers and together with the spiral form highlights how Mcm2-7 transitions through different conformational and assembly states as it matures into a functional helicase.

**\*For correspondence:**
alessandro.costa@cancer.org.uk (AC); mbotchan@berkeley.edu (MRB); jberge29@jhmi.edu (JMB)

**Present address:** †Centre For Ultra Structural Imaging, King's College London, London, United Kingdom

**Reviewing editor**: Stephen C Kowalczykowski, University of California, Davis, United States

## Introduction

The faithful copying of DNA requires the correct spatial and temporal assembly of replication machineries at specific chromosomal loci known as origins. In eukaryotes, origins are licensed for replication by recruitment of the Mcm2-7 complex, a ring-shaped helicase that serves as the principal unwinding activity for separating parental DNA strands (*Blow, 1993*; *Bochman and Schwacha, 2008*; *Costa et al., 2011*; *Lyubimov et al., 2012*). Mcm2-7 is initially loaded around duplex DNA as an inactive double hexamer by the origin recognition complex (ORC), Cdc6, and Cdt1 in the G1 phase of the cell cycle (*Evrin et al., 2009*; *Remus et al., 2009*), forming a stable intermediate known as the pre-replicative complex (pre-RC, *Diffley et al., 1994*; *Donovan et al., 1997*; *Maiorano et al., 2000*; *Sun et al., 2013*; *Yanagi et al., 2002*). Upon entry into S phase, Mcm2-7 associates with the GINS complex and Cdc45 generating an 11-member assembly termed the CMG (*Kanemaki et al., 2003*; *Gambus et al., 2006*; *Moyer et al., 2006*; *Pacek et al., 2006*; *Ilves et al., 2010*). GINS/Cdc45 assembly is dependent on the CDK kinase (*Zegerman and Diffley, 2007*), while post-translational modification of Mcm subunits 2, 4, and 6 by the Cdc7/Dbf4 kinase (DDK) further contributes to CMG activation (*Labib, 2010*; *Sheu and Stillman, 2010*). Following the assembly of replicative polymerases and replisomal scaffolding

**eLife digest** Before a cell divides, it must duplicate its DNA so that each new cell inherits its own copy of the genome. To do this, the DNA double helix must be unwound so that the two individual strands of DNA can serve as templates for making new DNA molecules. Unwinding begins when two helicase complexes, termed the Mcm2-7 rings, are loaded together onto the DNA.

At first, the two Mcm2-7 rings encircle the double-stranded DNA and remain bound together in an inactive form. Activating the Mcm2-7 rings requires the binding of five other proteins to each ring, which forms two larger complexes called CMG helicases. When the CMG helicases form, the two DNA strands separate and an individual Mcm2-7 ring ends up encircling each of the single DNA strands. However, how an activated CMG complex is assembled, and how it binds to and unwinds DNA, is not fully understood.

Now, Costa et al. have determined the three-dimensional structure of the fruit fly CMG helicase bound to a DNA double helix with a single-stranded overhang at one end. The activated Mcm2-7 ring binds to the overhang, which confirms previous findings indicating that the activated helicase prefers single-stranded over double-stranded DNA. The structure also shows that, as a CMG helicase slides along the single-stranded DNA towards the double-stranded DNA, it is the ring complex's 'motor domains' that lead the way, while its DNA-binding domains trail behind.

Costa et al. also found that disrupting some of the interactions between two of the five proteins that bind to the Mcm2-7 ring either prevented the replicative helicase from forming or made it unstable. Furthermore, it was revealed that one of these two proteins—called Cdc45—was ideally placed to capture the strand of DNA that might be accidentally released from the Mcm2-7 ring. It was also discovered that when the complex is bound to DNA, the motor domains of the Mcm2-7 complex change shape from a flat ring to a spiral structure; the DNA-binding domains, however, remain in a flat ring. Costa et al. note that this structure is similar to that adopted by many viral and bacterial helicases, and that it even shares many features with the molecular machinery that breaks down unneeded or damaged proteins inside cells.

Finally, Costa et al. were able to image a structure composed of two CMG complexes bound together. This reveals the relative orientation of the two Mcm2-7 rings before they separate and move in opposite directions to unravel the DNA. The findings of Costa et al., combined with previous structural work in this field, demonstrate that the Mcm2-7 helicase complex can adopt many different shapes as it is assembled on DNA and activated to support DNA replication.

factors (*Gambus et al., 2009*; *Muramatsu et al., 2010*), the two CMG particles split apart into discrete complexes that have been proposed to each encircle a single DNA strand during translocation (*Yardimci et al., 2010*; *Boos et al., 2012*).

At present, multiple aspects of the Mcm2-7 loading and activation cycle remain poorly understood. Although the six homologous subunits of one Mcm2-7 complex are known to pair with a second Mcm2-7 complex through their N-terminal domains in the context of a double-hexamer (*Evrin et al., 2009*; *Remus et al., 2009*), the precise register by which these subunits interact with each other across the two rings is not known. How DDK phosphorylation of the Mcm2, Mcm6, and Mcm4 N-termini (*Labib, 2010*), or how a DDK-bypass mutation in the N-terminus of either Mcm4 (*Sheu and Stillman, 2010*) or Mcm5 (*Jackson et al., 1993*), might aid in the switch from an inactive Mcm2-7 double hexamer state to a functional CMG is similarly unclear, particularly as Mcm4 is spatially segregated from Mcm2 and Mcm5 (*Costa et al., 2011*).

During unwinding and fork progression, the CMG translocates 3′→5′ along DNA. How the various components of the CMG engage nucleic acid strands during this process has remained ill-defined. Cdc45 has recently been shown to contain a RecJ exonuclease domain that can bind DNA but that is catalytically inactive (Petojevic et al., unpublished data, as well as *Sanchez-Pulido and Ponting, 2011*; *Krastanova et al., 2012*; *Szambowska et al., 2014*). Whether or how the Cdc45 RecJ fold might bind single DNA strands formed in the context of the CMG has not been established. Conflicting models likewise exist for how Mcm2-7 engages substrate DNAs as it moves 3′→5′ during strand separation, with biochemical data from archaeal MCMs and phylogenetic relationships to superfamily III (SFIII)

helicases (such as the SV40 Large T antigen and the papillomavirus E1 protein) predicting mutually exclusive binding orientations (*McGeoch et al., 2005*; *Enemark and Joshua-Tor, 2006*; *Rothenberg et al., 2007*; *Lee et al., 2014*).

To begin to understand several extant questions surrounding how the CMG is formed and operates at molecular level, we have determined structure of the full-length complex from *Drosophila melanogaster* in the presence of a 3′-tailed DNA duplex and the non-hydrolyzable ATP analog, ATPγS, using negative-stain electron microscopy and single-particle reconstruction methods. The structure establishes that: 1) the CMG preferentially associates with single-stranded DNAs over double-stranded substrates, 2) the C-terminal ATPase domains of Mcm2-7 form the leading edge of the motor as it advances on a duplex, and 3) the RecJ domain of Cdc45 is oriented to favor the capture of DNA segments that might accidently escape the Mcm2-7 pore. Comparison of the new structure with a previously-determined apo CMG model (*Costa et al., 2011*) shows that the Mcm2-7 ATPase domains of the complex transition from a planar, open ring into a closed, right-handed spiral in the presence of both DNA and nucleotide. Analysis of this state alongside other ring-ATPases shows that the MCM spiral is most similar to that adopted by the bacterial DnaB helicase upon engaging single-stranded DNA (*Itsathitphaisarn et al., 2012*), and that the GINS•Cdc45 complex bridges the junction between the ends of the spiral in a manner similar to that by which the Rpn1 accessory subunit spans a spiral Rpt1-6 ATPase assembly in the eukaryotic proteasome (*Lander et al., 2012*). Interestingly, examination of a subpopulation of CMG dimers present in our EM data shows how two Mcm2-7 complexes associate within a double hexamer and suggests that this dimerized state persists during CMG formation, prior to separation during fork progression (*Ilves et al., 2010*; *Yardimci et al., 2010*). Collectively, our observations establish that Mcm2-7 unwinds DNA using an approach distinct from that of superfamily III helicases and highlight several new Mcm2-7 ring configurations and assembly states accessed by the motor during the initiation of DNA replication.

## Results and discussion

### Determination of a higher-resolution CMG model

In a previous study, we determined the medium-resolution (28 Å) structures of the *Drosophila melanogaster* CMG helicase in both an apo state and bound to a non-hydrolyzable ATP analog (*Costa et al., 2011*). Though sufficient for mapping individual subunits within the CMG, both models revealed a planar structure for Mcm2-7, with GINS and Cdc45 spanning a gap that appeared between Mcm2 and Mcm5 when nucleotide was omitted. Since insights into where DNA might bind to the CMG or how binding might potentially alter the structure of complex were unclear, we set out to trap and image a prospective translocation intermediate of the CMG using 3D single-particle electron microscopy. A purified solution of the CMG was first mixed with a 20 bp duplex DNA substrate bearing a single-stranded 3′-dT$_{(40)}$ tail and passed over a sizing column in the presence of the non-hydrolyzable ATP analog, ATPγS, to form a ternary complex. The complex did not behave well during cryo-preservation attempts using holey-carbon EM grids, so samples were instead deposited onto continuous carbon grids and exposed to uranyl formate for negative staining. A total of 29,913 particles were selected from EM micrographs acquired with JADAS automated data collection software (JEOL, *Zhang et al., 2009*) on a JEM2100 electron microscope. Following particle picking and 2D averaging, a 3D model was generated by projection matching using a low-pass filtered (60 Å), free-hand test-validated, nucleotide-bound structure of the CMG as a starting model ('Materials and methods'; *Rosenthal and Henderson, 2003*; *Lyubimov et al., 2012*).

CMG particles imaged with DNA and ATPγS turned out to be quite uniform, permitting structure determination to a higher resolution than that obtained previously (18 Å vs 28 Å resolution, *Figure 1—figure supplement 1*). The resultant model (*Figure 1A*) in turn allowed for a more accurate fitting of the Mcm2-7 and GINS subunits (*Figure 1B,C*), revealing several new features. For instance, the location of Psf1 C-terminus, which was previously not visible, was now clearly evident, and could be readily fit to a recently-published full-length structure of Psf1 from an archaeal ortholog (*Oyama et al., 2011*; *Figure 1C*). Flexing within the Mcm2-7 ring was also apparent with the C-terminal lobes of different MCM subunits displaying markedly distinct degrees of movement with respect to their associated N-terminal regions (*Figure 1D,E*). Asymmetric positioning between the two tiers of an MCM ring has not been reported previously, demonstrating that these elements are conformationally independent of each other to some extent in the presence of DNA substrates.

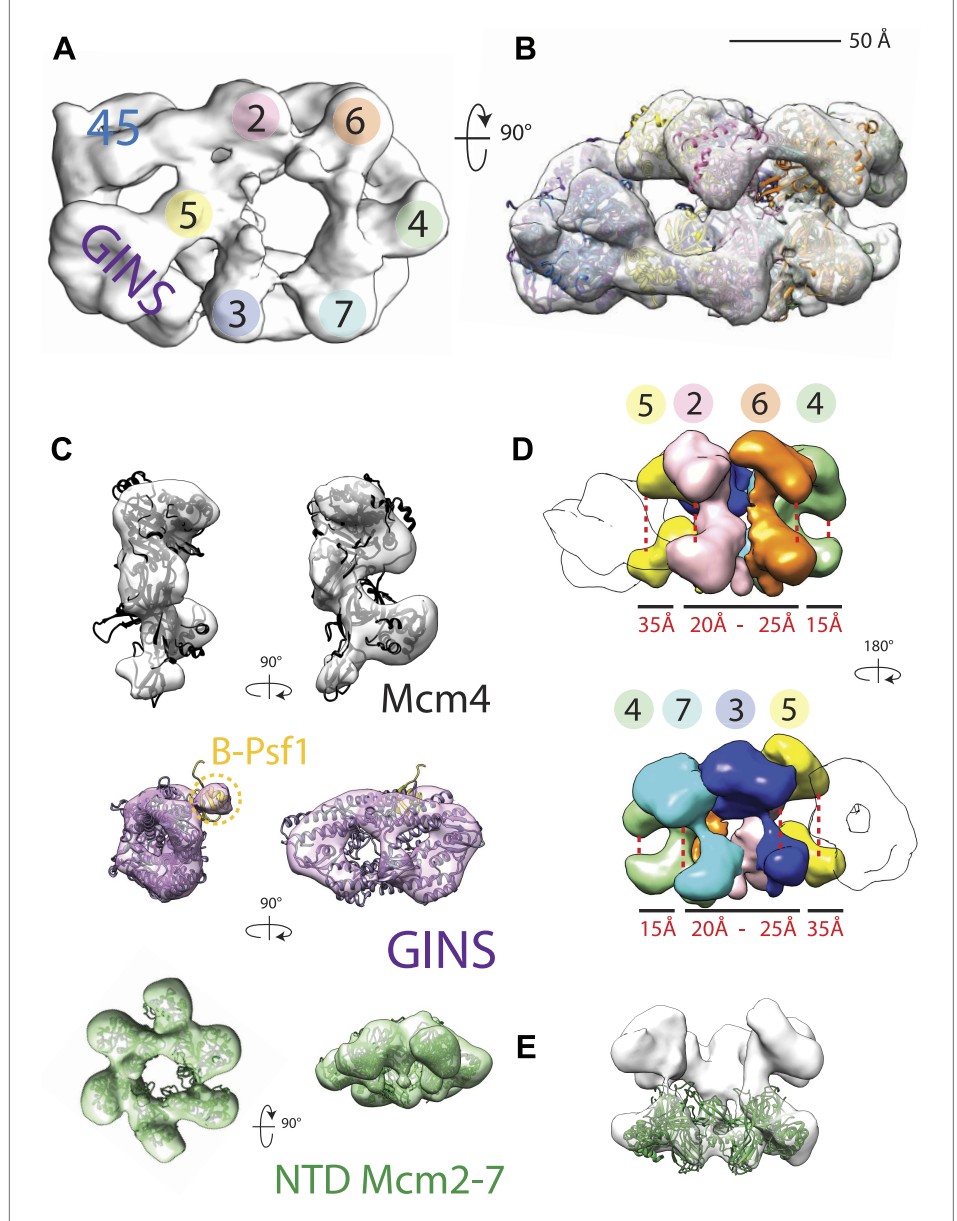

**Figure 1**. 18 Å resolution of a CMG–DNA–ATPγS complex. (**A**) Top-down view (N-terminal MCM face) of the CMG highlighting subunit positions. (**B**) Docking of homology models into the assembly. (**C**) Docked structures into segmented density for: *top*—a near-full-length, archaeal MCM monomer Mcm4 (PDB ID 3F9V); *middle*—the GINS complex (PDB ID 2Q9Q and 3ANW, 'Materials and methods'); *bottom*—the archaeal Mcm N-terminal domain hexamer (PDB ID 1LTL and 2VL6, 'Materials and methods'). (**D**) The N- and C-terminal domains of Mcm2-7 (colors) differentially flex around the helicase ring, with GINS–Cdc45 (white) wedging open Mcm5 in particular. (**E**) The N-terminal domains of Mcm2-7 are relatively planar, and are fit best by a hexameric, DNA-free structure of the archaeal MCM NTDs, indicating the observed intra-subunit flexing derives from ATPase domain movement.

The following figure supplement is available for figure 1:

**Figure supplement 1**. Overview of EM data.

## The CMG advances on duplex DNA from the C-terminal, ATPase side of Mcm2-7

The increase in resolution obtained for the CMG in the presence of DNA provided initial, support evidence for nucleic acid binding to the complex. More concrete evidence for DNA association

was apparent in electron density maps generated for the CMG, which showed a rod-shaped feature jutting away from the C-terminal face of Mcm subunits 2 and 5 (*Figure 2A*)—this feature is absent in DNA-free 3D reconstructions of the CMG (*Costa et al., 2011*). Because negative-stains are non-ideal for visualizing nucleic acids (*Grob et al., 2012*), we further assessed DNA binding by biotin-labeling the duplex end of the oligonucleotide, mixing the CMG–DNA samples with streptavidin, and collecting new single-particle EM data. Inspection of the resultant 2D class averages from this approach revealed clear additional density compared to the unlabeled CMG–DNA particles (*Figure 2B*), demonstrating that the tailed substrate indeed associates with the complex particles. Given the electron density features seen for the DNA and the distance the streptavidin 'pointer' resides from the complex, the EM data show that the CMG binds to the single-stranded end of the 3'-tailed DNA substrate, corroborating biochemical data indicating that the complex preferentially associates with and translocates along single-stranded DNA over duplex substrates (*Ilves et al., 2010*; *Fu et al., 2011*).

The ability to visualize not only DNA binding to the CMG but also the position of the duplex end with respect to the particle, resolves a key question concerning the polarity by which MCM helicases engage a presumptive translocation strand. MCMs and viral SF3 helicases, such as SV40 LTag and the papilloma virus E1 protein, are both AAA+ ATPases (*Neuwald et al., 1999*). This relationship, coupled with shared ability of MCMs and SF3 enzymes to translocate along DNA in a 3'→5' direction (*Kelman et al., 1999*; *Chong et al., 2000*; *Bochman and Schwacha, 2008*; *Moyer et al., 2006*), has suggested that members of two helicase families might operate by a common translocation mechanism. However, studies of E1 and archaeal MCMs bound to DNA substrates have yielded conflicting data concerning the direction by which DNA threads through the helicase pore. In E1, the 3' end of DNA has been observed by X-ray crystallography to lie proximal to the C-terminal motor domains of the helicase (*Enemark and Joshua-Tor, 2006*). By contrast, based on FRET measurements between a dye-labeled DNA/MCM pair, the converse has been reported for *Sulfolobus solfataricus* MCM (*McGeoch et al., 2005*; *Rothenberg et al., 2007*). In the new CMG structure, the streptavidin appended to the duplex DNA end can be clearly seen to localize next to the C-terminal, AAA+ domain face of the particle (*Figure 2B*). This finding not only demonstrates that a DNA segment bound by an MCM runs from the N-terminal collar to the ATPase motor region in a 3' to 5' direction, but also indicates that MCM and SF3 helicases bind substrates with opposing polarities.

## The RecJ domain of Cdc45 is poised to assist in the capture of DNA that might escape the Mcm2/5 gate

When the structure of the CMG was first reported, the fold of the associated Cdc45 subunit was unknown. As a consequence, although the general location of Cdc45 could be identified in both apo and ATP-bound forms of the CMG, the orientation and role of this subunit was left unresolved (*Costa et al., 2011*). Recently, however, the N-terminus of Cdc45 was shown to belong to the RecJ family of ssDNA exonucleases (*Sanchez-Pulido and Ponting, 2011*). Interestingly, within this grouping, Cdc45 belongs to an offshoot branch that can still bind DNA, but that also possesses natural amino acid substitutions which would appear to inactivate any native hydrolase functions (*Krastanova et al., 2012*).

To understand how the RecJ fold of Cdc45 interfaces with Mcm2-7 and GINS, we built a homology model for DmCdc45 based on *Thermus thermophilus* RecJ and docked it into the higher-resolution, DNA-bound CMG reconstruction. The catalytic core and DNA tracking domain of the homology model fit unambiguously into only one region of the Cdc45 density (*Figure 3A*), leaving only a single, unaccounted for region (most likely corresponding to the C-terminal segment of Cdc45 outside the defunct exonuclease core, or possibly to the N-terminal extension present in Mcm2) that interdigitates between the N-terminal 'A-domains' of Mcm5 and Mcm2 (*Figure 3B*, *Figure 3—figure supplement 1*). Notably, in placing the Cdc45 RecJ domain, we found that this element appeared to contact the now-apparent C-terminal 'B-domain' of Psf1 (*Figure 3A*). To test whether this interaction might be real or fortuitous, we subjected the Psf1 B-domain surface to site-directed mutagenesis and tested the ability of the mutant subunits to support binding to both Mcm2-7 and Cdc45 ('Materials and methods'). Ablation of either the Psf1 B-domain region (residues 185–202) or Cdc45 N-terminal region (residues 1–99) prevented CMG formation under the conditions used to purify the intact assembly (*Figure 3—figure supplement 2*). Likewise, while point mutations were unable to disrupt CMG formation as judged by co-immunoprecipitation, a quadruple Psf1 mutant (E190A/L192A/V193A/R194A) proved unable to interact with Cdc45. Together, these data indicate that the Cdc45–Psf1 interaction evident from the EM

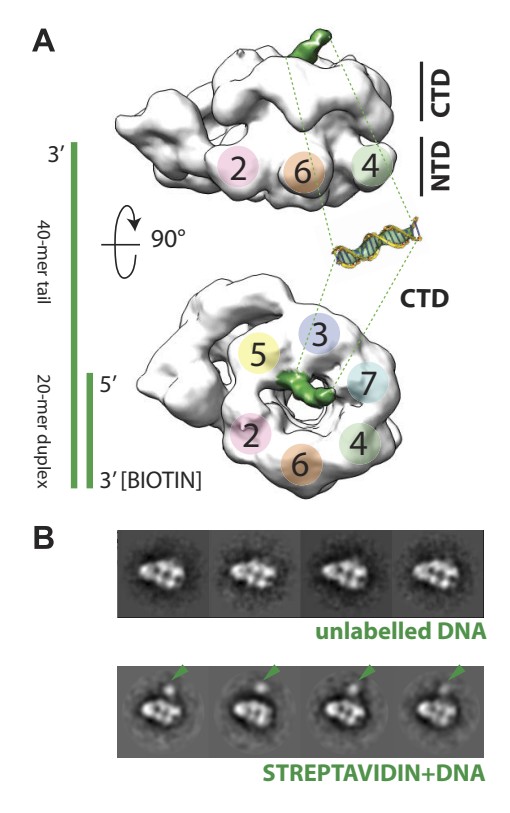

**Figure 2**. Polarity of DNA binding by the CMG. (**A**) Observed experimental density seen at low contours reveals a rod-shaped extension (green) of comparable length to that expected for a 20mer DNA duplex that extends from the Mcm C-terminal motor domain. This feature is absent in DNA-free CMG reconstructions (*Costa et al., 2011*). A schematic of the relative single- and double-stranded DNA regions of the substrate used for the present studies is shown at left. (**B**) Comparison of DNA–CMG class averages with and without streptavidin-labeling clearly marks the duplex end of the 3'-tailed duplex substrate. Note how the streptavidin density sits at a distance from the body of the CMG, indicating that the majority of the duplex region of the substrate is not bound by the CMG.

data plays a critical role in CMG formation and/or stability.

In previous apo and ATP-bound models of the CMG, the particle was seen to transition from a conformation in which the Mcm2/5 interface was open to one in which it was closed (*Costa et al., 2011*). This transition in turn pinched off the large single channel that ran through the particle into two smaller channels, sealing the interior of Mcm2-7 away from the inner surface of GINS–Cdc45. In the DNA-bound model, the CMG still exhibits two channels; however, docking of the Cdc45 RecJ domain shows that its exonuclease/DNA-tracking groove is offset by 90° with respect to the central axis of the Mcm2-7 pore (*Figure 3C*). This orientation indicates that, were Cdc45 to bind DNA in a manner similar to RecJ, it would be poised to capture the leading DNA strand that might escape from Mcm5-2 gate (*Figure 3D*). Consistent with this idea, cross-linking data in work to be published elsewhere (Petojevic et al.) show both that Cdc45 engages the leading strand of a fork substrate only in the absence of nucleotide and that this interaction is ablated by the mutation of residues suggested by the model to be important for DNA binding.

## DNA and nucleotide remodel the Mcm2-7 AAA+ ATPase subunits into a right-handed spiral

How ATP-dependent physical movements within hexameric helicases are coupled to DNA binding and unwinding has long been a central question in the field (*Singleton et al., 2007*; *Enemark and Joshua-Tor, 2008*; *Lyubimov et al., 2011*). Notably, when comparing the new DNA-bound CMG model to the prior substrate-free state, we found that the AAA+ ATPase ring is no longer flat, but instead adopts a clear right-handed spiral (*Figure 4A,B*). This change in conformation does not propagate into the N-terminal domains, which maintain a roughly planar character (*Figure 1E*), but is instead offset by the variable flexing seen for the C-terminal domains in different positions around the ring (*Figure 1D*). The observed asymmetry between the two MCM tiers indicates that the N-terminal domains form a relatively stable collar that likely helps to coordinate and restrain movements of the associated C-terminal ATPase regions.

Closer analysis of the AAA+ spiral reveals several features that have important implications for the action of MCM subunits during DNA unwinding. First, the largest inter-subunit shifts within the Mcm2-7 ring, which occur between Mcm subunits 2 and 5, also correspond to the point where the GINS–Cdc45 complex docks against the helicase. Inspection of the DNA-bound model reveals that GINS–Cdc45 does not simply straddle the Mcm2/5 interface, but that portions of the accessory subunits actually wedge themselves between the N- and C-terminal tiers of the Mcm2-7 ring (*Figure 1D*). This action widens the exterior groove between the MCM N- and C-terminal domains at their points-of-contact with GINS–Cdc45, and is offset by a concomitant narrowing of the groove on the exterior MCM face opposite the GINS–Cdc45 binding site (i.e., Mcm4, *Figure 1D*). The structural consequences resulting

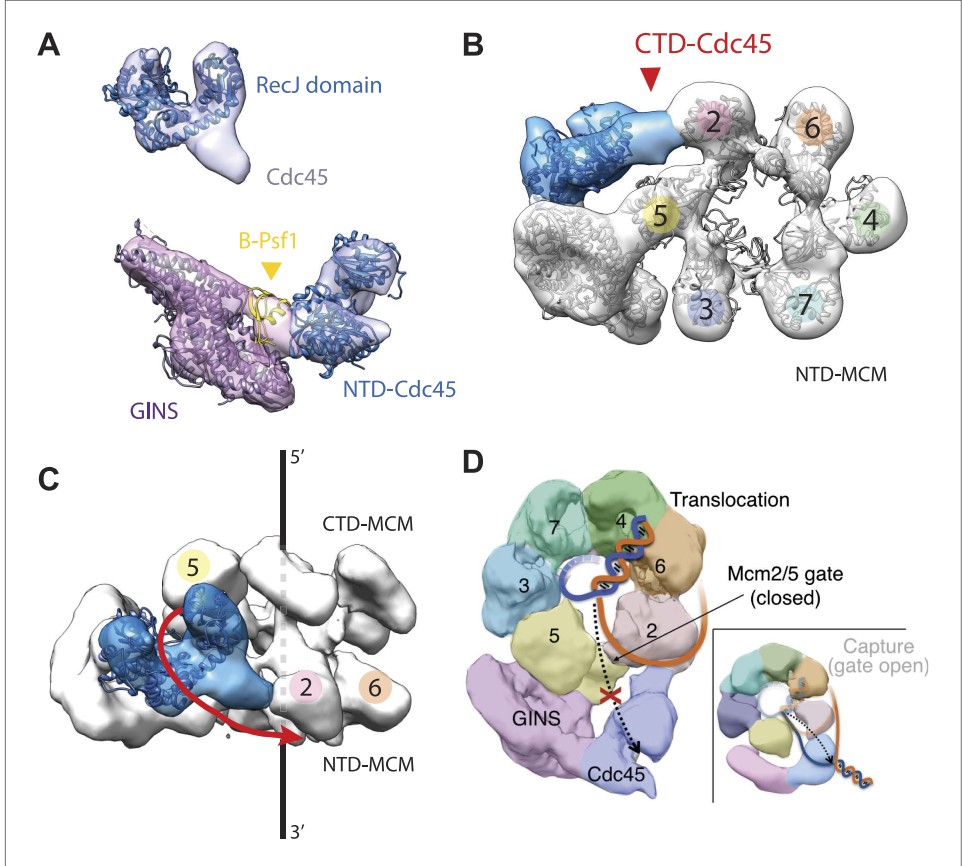

**Figure 3**. Cdc45 is positioned to permit trapping of single-stranded DNA. (**A**) *Top*—Segmented electron density corresponding to Cdc45. A prominent horseshoe-shaped region fits well to the catalytic core of the homologous RecJ exonuclease (PDB ID 1IR6). *Bottom*—Docked models of RecJ and full-length GINS (generated using an archaeal Psf1 homolog, PDB IDs 2Q9Q and 3ANW) into DNA-bound CMG reconstructions highlight a previously unobserved interaction between the B-domain of Psf1 and the exonuclease-like domain of Cdc45. (**B**) An extension of the Cdc45 RecJ-like region contacts and interdigitates between the A-domains of the Mcm2 and Mcm5 N-terminal regions. (**C**) The Mcm2-7 central channel (black line) and the Cdc45 DNA tracking groove (red arrow) are offset by ~90°. (**D**) Schematic showing how the single-stranded DNA-binding groove of the Cdc45 RecJ-like domain could facilitate the capture of a leading strand segment if the Mcm5-2 DNA gate were to transiently open.

The following figure supplements are available for figure 3:

**Figure supplement 1**. Comparison between the open N-terminal domain of Drosophila Mcm2-7 and the closed N-terminal region of the CMG.

**Figure supplement 2**. The Psf1 C-terminus and Cdc45 N-terminus are critical for the CMG formation.

from GINS–Cdc45 binding suggests that these accessory subunits not only play a role in blocking access through the Mcm2/5 gate, as has been seen previously (*Costa et al., 2011*), but that they also help stabilize a spiral configuration of ATPase centers when DNA is present. Since the mutation of active site residues at the Mcm2/5 interface ablates helicase activity (*Bochman et al., 2008*; *Ilves et al., 2010*), it is likely that the spiral state observed here, which positionally offsets the ATP-binding site of Mcm5 from the arginine-finger residue of Mcm2, inter-converts with another conformation in which the Mcm2/5 interface is remodeled to form a catalytically functional ATPase center during the translocation cycle. Hence, a need for GINS–Cdc45 in preventing DNA from escaping Mcm2-7 would likely be infrequent and limited to instances when the Mcm2/5 gate accidentally opens for an extended period of time, such as at a roadblock created by other nucleoprotein complexes or DNA damage.

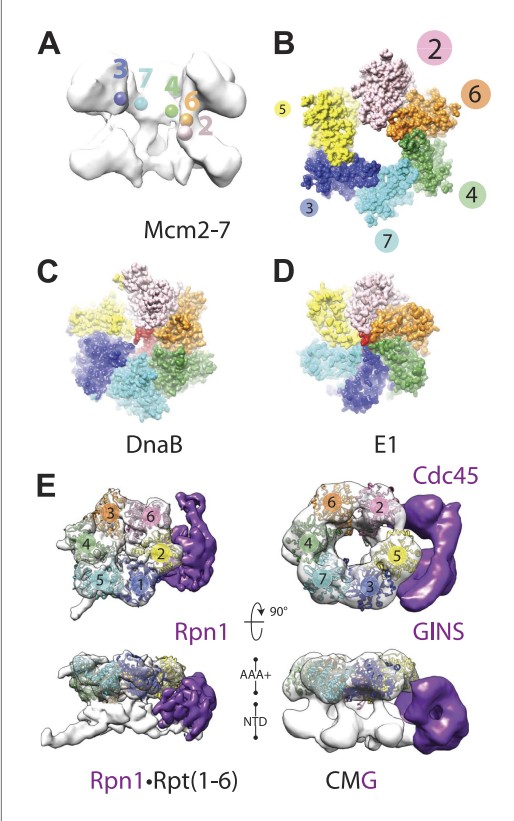

**Figure 4**. Global comparison of the DNA-bound Mcm2-7 region of the CMG with other hexameric helicases and ATPases. (**A**) Cut-away view (removing Mcm5) of the Mcm2-7 central channel highlights a spiral organization for the Mcm2-6-4-7-3 AAA+ ATPase regions. Colored spheres demarcate the approximate center of mass for AAA+ pore loops as derived from the docking of MCM AAA+ domain as shown in *Figure 1B*. (**B**) Top-down view (from the N-terminal face) of MCM AAA+ domains docked into the DNA-bound CMG reconstruction showing the existence of a right-handed spiral. The CMG density has been removed for clarity. (**C**) In the presence of a single-stranded DNA, bacterial DnaB can adopt a right-handed spiral with a moderately-wide pore (PDB ID 4ESV, *Itsathitphaisarn et al., 2012*). (**D**) The E1 helicase assembles into a right-handed spiral with a relatively narrow pore (PDB ID 2GXA, *Enemark and Joshua-Tor, 2006*). (**E**) Comparison of the AAA+ ring of the eukaryotic proteasome with Mcm2-7 region of the DNA- and ATPγS-bound CMG. The non-ATPase subunit Rpn1 binds to the side of the Rpt1-6 hetero-hexamer, wedging itself between the N-terminal and C-terminal tiers of the ATPase ring and helping to promote the formation of a right-handed ATPase domain spiral. Similar architectural features are apparent within the DNA–ATPγS–CMG complex, where GINS–Cdc45 occupy an analogous position.

A second unexpected feature of the DNA-bound CMG complex is that the spiral is more pronounced than that seen in SF3 helicases, and instead more closely approximating the spiral evinced by a RecA-family helicase, DnaB, in the presence of DNA (*Figure 4B–D*). The width of the Mcm2-7 central channel (as measured from homology models of the motor domains docked into the EM density) is likewise significantly larger (~30–35 Å) compared to E1 (~14 Å), and more closely approaches that of DnaB (~22 Å). Interestingly, in E1 and DnaB the difference in channel diameter and subunit rise between the two proteins sculpts the DNA substrate bound by each helicase into a single-stranded helix whose relative pitches differ significantly; these geometric differences allow each subunit of DnaB to engage two nucleotides of DNA (*Itsathitphaisarn et al., 2012*), whereas E1 binds only a single nucleotide per protomer (*Enemark and Joshua-Tor, 2006*). The similarity of the Mcm2-7 spiral to DnaB raises the interesting possibility that the helicase might translocate with a step-size greater than one nucleotide per ATP consumed; consistent with this notion, a recent study has shown that the MCM N-terminal DNA-binding collar of *Pyrococcus furiosus* binds four nucleotides per subunit (*Froelich et al., 2014*).

A third notable attribute of the ternary DNA–CMG–ATPγS model is that several structural features of the complex turn out to be most similar not to replicative helicases, but to a completely orthogonal system, namely, the regulatory subcomplex of the eukaryotic proteasome. The proteasome consists of several discrete subcomplexes including a heterohexameric unfoldase region, termed the 'base', which (like the CMG) contains six homologous AAA+ ATPase subunits (Rpt1-6) (*Forster et al., 2013*). Recent cryo-EM studies have imaged the complete 26S yeast proteasome bound to ATP at ~9 Å resolution showing that the AAA+ subunits of the base also form a right-handed spiral (*Lander et al., 2012*). Comparison of proteasome spiral with that seen here for the CMG shows that these regions of the two systems exhibit a surprisingly similar global architecture (*Figure 4E*). Moreover, the proteasome also contains an accessory subunit (Rpn1) that—as observed here for GINS–Cdc45 in the context of the CMG—wedges itself between a subset of ATPase and OB-fold domains present in Rpt1-6 (*Lander et al., 2012*; *Figure 4E*). The structural congruencies exhibited between the CMG and proteasome ATPase subcomplexes suggest that, even though the substrates for the two systems differ greatly, both motors may

share certain commonalities in how ATP turnover is coupled to movements that promote translocation. Such a similarity could underlie both the pronounced asymmetry of the CMG and proteasome ATPase rings, and the relatively high degree of tolerance shown by both systems toward active-site mutations within certain subunits (*Moreau et al., 2007*; *Ilves et al., 2010*; *Beckwith et al., 2013*).

## Identification of a dimeric CMG configuration

Although the CMG has been observed to operate as a discrete single complex during replication (*Yardimci et al., 2010*), the loading of the Mcm2-7 hexamer onto DNA by ORC, Cdc6, and Cdt1 during initiation results in the transient formation of a catalytically inactive, head-to-head double hexamer intermediate (*Evrin et al., 2009*; *Remus et al., 2009*). The MCM N-terminal domains have been shown to comprise the dimer interface of the double hexamer (*Fletcher et al., 2003*; *Costa et al., 2006*; *Remus et al., 2009*), and create a critical target site for the Dbf4-dependent Cdc7 protein kinase DDK (*Labib, 2010*; *Sheu and Stillman, 2006*, *2010*); phosphorylation of the N-terminal tails of Mcm2/4/6 alters the configuration of the Mcm2-7 complex, but does not directly promote Mcm2-7 dissociation (*On et al., 2014*).

At present, it is unclear how the Mcm2, Mcm4, and Mcm6 subunits are aligned with respect to one another in the Mcm2-7 double hexamer. However, in the course of our studies, we found a new configuration of the CMG that sheds light on this issue. In particular, a small (~5%) but consistent population of CMG particles was seen to form a clear dimeric species that adopts a distinctive head-to-head configuration through its MCM N-terminal regions, and which consistently orients the GINS–Cdc45 subcomplex toward opposing sides of the two ring (*Figure 5A*). The abundance and uniformity of CMG dimers, which were noted in several independent preparations, suggests that these particles represent a naturally occurring state of the assembly.

The observed organization of the CMG dimers has several implications for the formation and function of the helicase. For example, the 180° offset of GINS and Cdc45 present in the dimer places each Mcm2/5 gate on the opposite sides of the complex, facing away from each other (*Figure 5B,C*). During origin melting, this configuration would allow a single DNA strand to escape each Mcm2-7 hexamer without steric interference from its partner CMG, enabling particle separation and the formation of two independent replication forks. The organization of the CMG dimer also indicates that the N-terminal regions of Mcm2 and Mcm5 of one hexamer associate in *trans* with the N-terminal regions of Mcm6 and Mcm4 of the partner hexamer. Such an interaction would help explain why Mcm4 and GINS have been seen to interact in pulldown studies (*Ilves et al., 2010*), even though the two factors map to distal positions of the CMG in the context of a monomer (*Costa et al., 2011*). Finally, the observed arrangement suggests that the ability of DDK to activate Mcm2-7 by phosphorylation of Mcm4, 2, and 6 (*Labib, 2010*) (as well as the ability of the Bob1 mutation in Mcm5 to bypass the requirement for DDK [*Jackson et al., 1993*]) could result from a destabilization of the CMG dimer contacts that are symmetrically apposed across the N-terminal collar (*Figure 5C*). Altogether, our data establish that the two Mcm2/5 gates of a double hexamer are spatially segregated from each other and indicate that separation of Mcm2-7 double hexamers occurs subsequent to CMG formation. Such a mechanism is consistent with recent findings showing that the phosphorylation by DDK is insufficient to promote the separation of Mcm2-7 double hexamers on its own (*On et al., 2014*).

The existence of a dimeric CMG state, coupled with prior views of Mcm2-7 and the DNA-bound conformation of the Mcm2-7 ATPase domains seen here, highlights the innate plasticity of the MCM ring and the means by which different factors help remodel the helicase to support appropriate loading and activation during the initiation of DNA replication (*Figure 6*). Biochemical studies using *Saccharomyces cerevisiae* proteins first showed that Mcm2-7 on its own possesses a natural discontinuity—the 'Mcm2/5 gate'—through which DNA can enter and exit the helicase pore (*Bochman and Schwacha, 2007*). Structural studies have corroborated this observation, additionally showing that metazoan Mcm2-7 rings preferentially assume a left-handed lock-washer shape and that ATP alone is incapable of fully inducing ring closure (*Costa et al., 2011*; *Lyubimov et al., 2012*). Following the action of ORC, Cdc6, and Cdt1, two Mcm2-7 hexamers become locked into a planarized co-joined ring (*Evrin et al., 2009*; *Remus et al., 2009*); in the presence of GINS, Cdc45, and single-stranded DNA, the Mcm2-7 ATPase domains shift again, but now into a right-handed spiral conformation (*Figure 4*). Thus, Mcm2-7 undergoes a chiral-flip in architectural state as it matures into a functional helicase with loading, activation, and DNA binding all appearing to participate in these transitions.

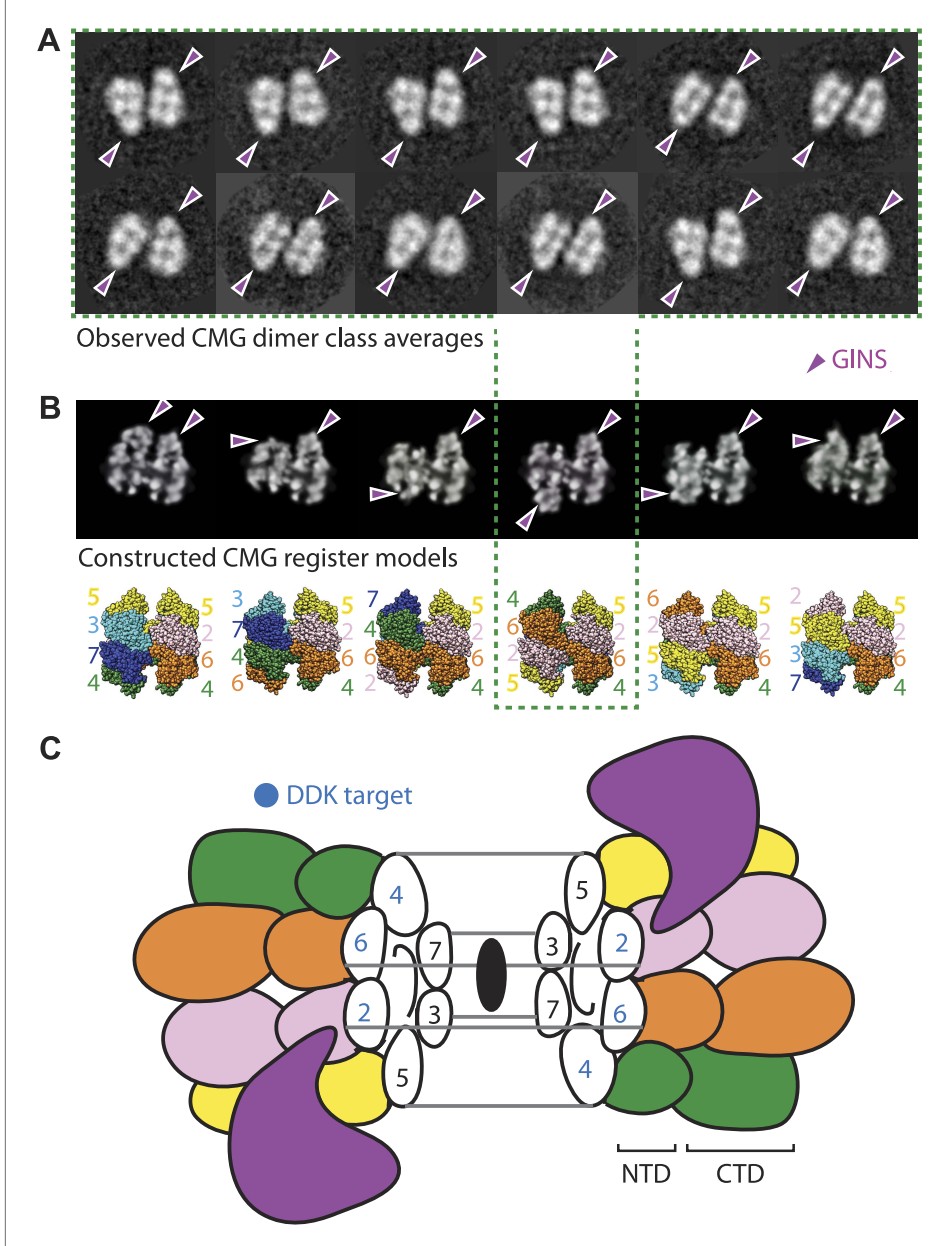

Figure 5. The CMG can form head-to-head dimers that establish the interactions between Mcm2-7 double hexamers. (**A**) Experimentally observed 2D class averages showing that the CMG forms a defined double-hexamer in which the GINS-Cdc45 subcomplex is rotationally offset toward opposite sides of the Mcm2-7 rings. Arrowheads mark the position of GINS/Cdc45. The structure of the archaeal N-terminal double hexamer is shown to depict the six possible Mcm2-7 registers that were tested. The class averages show that the interface between the N-terminal collars can partially crack open in some instances, suggesting that this region is somewhat unstable when the CMG is bound to ssDNA. (**B**) Computationally derived 2D projections of double-hexameric 3D CMG models in which Mcm2-7 N-terminal domain dimers (modeled on an archaeal Mcm crystal structure of this region, PDB ID 1LTL) have been manually offset by distinct rotational registers. The reference-free class averages best resemble a model in which Mcm5 from ring 1 juxtaposes with Mcm4 from ring 2, although our data cannot formally rule out a configuration where Mcm4 might juxtapose with Mcm3 or Mcm2 (although in these configurations the two Mcm2/5 gates would still remain misaligned). Arrowheads mark the position of GINS/Cdc45. (**C**) Cartoon representation of the Mcm2-7 double ring register formed in head-to-head dimers of the CMG.

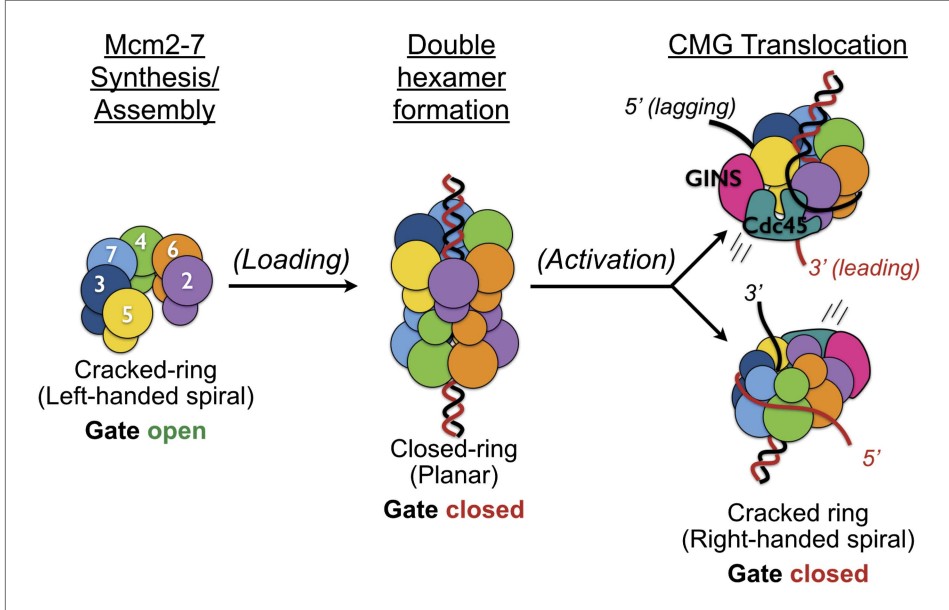

**Figure 6**. Overview of Mcm2-7 organization, remodeling, and gate status during initiation and CMG formation. Following synthesis and assembly, the isolated metazoan Mcm2-7 motor forms an inactive left-handed spiral, irrespective of nucleotide state, with a discontinuity between Mcm5 and Mcm2. The action of ORC, Cdc6 and Cdt1 results in the loading of a planar, head-to-head Mcm2-7 double hexamer onto a duplex DNA in which the Mcm2/5 gates are closed. Following loading, Dpb11-Sld2-Sld3 (11-2-3) chaperone GINS and Cdc45 onto the Mcm2-7 double hexamer and along with DDK promoted phosphorylation events, help promote both DNA melting, CMG formation and replication fork separation. Structural analysis of dimeric CMG particles (*Figure 5*) indicates that the Mcm2/5 gates are localized on the opposing sides of the dodecameric Mcm2-7 complex and that CMG formation precedes separation of the double hexamer.

Why might Mcm2-7 traverse through such disparate intermediates? One possibility is that Mcm2-7 initially assembles into an inactive conformation to prevent inadvertently triggering aberrant replication events. An alternative possibility, not necessarily incompatible with the first, is that Mcm2-7 might transition through different forms as a means to promote the melting of duplex replication origins. Interestingly, an important attribute of some of the states adopted by Mcm2-7 is the existence of a breach or subunit offset in the ATPase ring between the Mcm2 and Mcm5 subunits. Given the topological barrier that extended DNA segments present to encirclement by toroidal proteins, it is notable that two MCM-associated factors—Cdt1 and GINS–Cdc45—both appear to occlude the Mcm2/5 interface (*Costa et al., 2011*; *Sun et al., 2013*). Hence, controlled access through the Mcm2/5 gate appears useful not only for regulating DNA access into a preformed Mcm2-7 hexamer (*Bochman and Schwacha, 2007*), but also for preventing accidental DNA egress in the apo state and at stalled forks (Petojevic et al., unpublished data).

## Concluding remarks

The structure of the eukaryotic CMG complex bound to nucleotide and a 3'-tailed DNA substrate helps resolve many outstanding questions surrounding the mechanism by which this key replication assembly forms and operates. The structure reveals that the RecJ domain of Cdc45, which recent data have found to bind to escaped leading strand substrates (Petojevic et al., unpublished data), takes up a position on the CMG that would aid in capturing a DNA segment that might escape from the Mcm2-7 pore. Our EM data show that the nucleotide-loaded CMG orients on single-stranded DNA such that the 3' end enters first through the C-terminal AAA+ domains of Mcm2-7, consistent with prior biochemical data indicating that archaeal MCMs move along DNA with the opposite polarity as superfamily III helicases (*McGeoch and Bell, 2005*; *Rothenberg et al., 2007*). The binding of DNA to the CMG also induces the formation of a right-handed spiral in the MCM ATPase domains whose overall structure more closely mimics bacterial RecA-family replicative helicases than its viral AAA+

cousins, and whose relative arrangement of AAA+ domains and accessory subunits is mirrored by the regulatory subcomplex of the eukaryotic proteasome (*Lander et al., 2012*). Finally, we find that the CMG can form head-to-head dimers in which the two Mcm2/5 gates are fully offset from each other, helping to fill in key gaps concerning the higher-order organization of the Mcm2-7 double hexamer and how the CMG matures from this state into two single particles that encircle complementary single strands. Future studies will be needed to definitively establish the full paths of the leading- and lagging-strand DNA bound to the CMG and the extent to which the disparate AAA+ and RecA-family ring-translocases share or diverge in coupling ATP turnover to specific subunit movements that drive substrate translocation.

## Materials and methods

### Cloning and construction of baculoviruses

Baculoviruses were constructed as previously described (*Ilves et al., 2010*). Briefly, the MCM3 gene was tagged with FLAG epitope at the amino terminus through 5′ PCR oligonucleotides that inserted the epitope in frame. Mcm2, Mcm3, Mcm4, Mcm5, Psf1, Psf2, and Psf3 cDNAs were inserted between EcoRI and SpeI restriction sites of the pFastBac1 vector. The cDNA of Mcm6 and Mcm7 was inserted between BamHI and SpeI sites, and Cdc45 and Sld5 cDNA between EcoRI and Xba restriction sites. These vector templates were used for generation of CMG mutants through PCR based mutagenesis. Sequencing was used to verify the entire protein coding regions of all generated pFastBac1 constructs. Specific deletion mutants of either the Psf1 C-terminus or Cdc45 N-terminus included $Psf1^{1-139}$, $Psf1^{1-170}$, $Psf1^{1-176}$, $Psf1^{1-184}$, and $Cdc45^{\Delta1-99}$. C-terminal deletions were constructed by introducing stop codons after the desired residue and by removing any remaining C-terminal sequences present from the original cDNA clone. N-terminal deletions were constructed by the removal of the pertinent cDNA regions and by introducing a start (ATG) codon in front of the desired residue. The same restriction enzyme sites as described above were used to subclone all truncation constructs into the pFastBac1 vector. To target the Psf1–Cdc45 interface, alanine substitutions in the C-terminus of Psf1 were introduced individually or in parallel into for Glu190, Leu192, Val193, and Arg194 by site-directed mutagenesis.

### Nucleoprotein complex preparation

The CMG complex was purified as previously described (*Ilves et al., 2010*), but with the following changes. Briefly, after co-infection with 11 distinct baculoviruses, proteins were expressed for 72 hr in Hi5 cells (Invitrogen, Carlsbad, CA) by culturing in 500-ml spinner flasks and co-infecting at $1.2^10^6$ cells/ml density. Cells were lysed by hypotonic shock combined with one freeze–thaw cycle and Dounce homogenization. The lysate was clarified by centrifugation, and the protein complex was immunoaffinity purified over anti-flag (M2) antibody-conjugated agarose beads (Sigma, St. Louis, MO) to bind a FLAG-tag on Mcm3. To isolate the fully intact CMG from incomplete complexes, the protein preparation was passed over a Mono S HR 5/5 ion exchange column, followed by a Mono Q HR 5/5 ion exchange chromatography using an ÄKTA Purifier (GE Healthcare, Piscataway, NJ). Peak fractions were collected and then both further purified and concentrated using a Mono Q PC 1.6/5 column coupled to a Pharmacia SMART system. The purified material was dialyzed into a buffer containing 25 mM HEPES (pH 7.6), 50 mM sodium acetate, 10 mM magnesium acetate, and 1 mM DTT. The final protein concentration as measured by the Bradford protein assay was 1.2 mg/ml.

Oligonucleotides used for nucleoprotein complex reconstitution were synthesized by Integrated DNA Technology and shipped as lyophilized pellets. Oligo 'LEAD60' contained the sequence 5′–GGG-CAC-TTG-ATC-GGC-CAA-CCT-$T_{39}$–3′, while '3BTNLAG20' contained the sequence 5′–GGT-TGG-CCG-ATC-AAG-TGC-CC–biotin–3′. The oligonucleotides were dissolved in the CMG buffer and quantified by $A_{260}$. For annealing, LEAD60 and 3BTNLAG20 were mixed in equimolar amounts, briefly heated to 95°C and slow-cooled to 4°C. Previous work has shown that the ATPγS binds to a duplex-DNA substrate containing a 40mer 3′ single-stranded DNA tail (*Ilves et al., 2010*). To form helicase/DNA complexes, 10 nmol of concentrated CMG were mixed with the annealed DNA with a 1.2 molar excess of nucleic acid in CMG buffer plus 0.1 mM ATPγS. After incubation at room temperature for 30 min, the sample was passed over a Superose 6 PC 3.2/30 gel filtration column using an ETTAN micropurification system (GE Healthcare). A 50-μl fraction containing the center of the nucleoprotein elution peak was collected and immediately used for negative stain grid preparation, either with or without co-incubation with 1.2-fold molar excess of streptavidin.

## Negative stain grid preparation

EM grids were prepared by floating a thin layer of continuous carbon over a 400-mesh copper grid (Electron Microscopy Sciences, Hatfield, PA) using a custom-made carbon-floating device. Four microliters (~40 ng) of the CMG–DNA or CMG–DNA–Streptavidin complex were then applied onto freshly glow-discharged grids for 30 s. The grids were laid on top of 75-µl drops of a fresh 2% (wt/vol) uranyl formate solution and stirred for five consecutive 10-s staining steps. The staining solution was then blotted dry and the grids were stored.

## Electron microscopy

Nucleoprotein particles were imaged using a JEM-2100 LaB6 electron microscope (JEOL, Japan) operated at 200 kV. Images were recorded at a nominal magnification of 50,000× on a Ultrascan 4k × 4k CCD camera (Gatan, Pleasanton, CA), resulting in a 2.14 Å pixel size at the specimen level. The JEOL Automated Data Acquisition System (JADAS, *Zhang et al., 2009*) was used to automatically collect low-dose images with a 0.5 to 3.5 µm defocus at around 35 electrons per $Å^2$. In addition, manual data collection was performed under the same imaging conditions. In total, 579 micrographs were collected for the non-labelled and 436 micrographs for the labeled nucleoprotein complex.

## Image processing and atomic docking

Particles were semi-automatically picked and phase flipped using the EMAN2 package, version 2.05 (*Tang et al., 2007*). Reference free two-dimensional class-sums were obtained using RELION, version 1.2 (*Scheres, 2012*), except for the streptavidin-DNA-bound CMG complex that was processed using Imagic (*van Heel et al., 1996*) and the rotation and classification protocol implemented in *Costa et al. (2011)*. Working with the ADP(BeF3) bound CMG as a starting model (*Costa et al., 2011*), multi-model three-dimensional refinement was performed with an iterative projection-matching and back-projection protocol that employs libraries from the EMAN2 and SPARX software packages (*Hohn et al., 2007*; *Tang et al., 2007*) or, in a parallel effort, using three-dimensional classification and refinement routines as implemented in RELION (*Scheres, 2012*). Refinement of the starting models began using an angular increment of 15°, progressing down to 2° with EMAN2/SPARX, and default parameters in RELION 1.2 (7.5° angular sampling, 5 pixel search range, and 1 pixel search step). The two approaches yielded virtually identical results (*Figure 1—figure supplement 1*) with the best structure containing 7,409 particles (obtained with EMAN2/SPARX and shown in all figures). The resolution was estimated by the 'gold-standard' Fourier Shell Correlation approach implemented in RELION (*Scheres, 2012*). 3D-maps were segmented using the Segger program (*Pintilie and Chiu, 2012*) in UCSF Chimera (*Pettersen et al., 2004*). The Chimera 'Fit in Map' option was used for rigid-body fitting of crystal structures and to generate all surface renderings included in the figures (*Pettersen et al., 2004*). The full-length Psf1 protein was modelled by superposing the *Thermococcus kodakaraensis* GINS51 (PDB entry 3ANW) onto the human Psf1 structure. The N-terminal Mcm2-7 collar was modelled by superposing protomers of the *Methanothermobacter thermautotrophicus* MCM (PDB entry 1LTL) or the *Sulfolobus solfataricus* MCM (PDB entry 2VL6) onto the isolated oligomerization subdomain of the *M. thermautotrophicus* MCM. The two structures exhibit markedly distinct configurations in their N-terminal α-helical region (subdomain A), with the *S. solfataricus* MCM region matching best to the N-terminal domain configuration of Mcm4 and Mcm5. The atomic model of the near full-length *S. solfataricus* MCM (PDB entry 3F9V) was found to fit the entire Mcm4 electron density region, whereas slight rigid body transformations of the MCM AAA+ domain relative to the N-terminal region were allowed to best fit the electron density of Mcms 2, 3, 5, 6, and 7. A homology model based on the bacterial RecJ (PDB entry 1IR6) was used for docking of Cdc45. The EM map has been deposited in the 3D-EM database (www.emdatabank.org) with accession code EMD-2772.

## Acknowledgements

The authors thank Raffaella Carzaniga and Lucy Collinson for technical support, Gabriel Lander for sharing the atomic model of Rpt1-6, and the members of the Costa and Berger lab for useful comments. This work was supported by the Cancer Research UK (AC), a PhD fellowship from the Boehringer Ingelheim Fonds (to TP), the NRSA (GM821972 to JP), the NIGMS (GM071747 to JMB), and the NCI (CA R37-30490 to MRB).
    Author Contributions

AC, MRB, and JMB designed the research. JP and TP purified the CMG complexes and TP performed the pulldown assays. AC, JP, and II designed and prepared the nucleoprotein complexes. AC prepared all EM grids, and performed the 3D reconstruction/modeling work helped by LR. LR and AC collected the data with the support of RF and KMG; AC, LR, and PS performed the 2D image analysis. AC, MRB, and JMB analyzed the data and wrote the manuscript.

## Additional information

### Competing interests

MRB: Reviewing editor, *eLife*. The other authors declare that no competing interests exist.

### Funding

| Funder | Grant reference number | Author |
|---|---|---|
| Cancer Research UK | | Alessandro Costa |
| National Cancer Institute | | Michael R Botchan |
| National Institute of General Medical Sciences | | James M Berger |
| National Institutes of Health | National Research Service Award, GM821972 | James J Pesavento |
| Boehringer Ingelheim Fonds | | Tatjana Petojevic |

The funders had no role in study design, data collection and interpretation, or the decision to submit the work for publication.

### Author contributions

AC, Conception and design, Acquisition of data, Analysis and interpretation of data, Drafting or revising the article, Contributed unpublished essential data or reagents; LR, Acquisition of data, Analysis and interpretation of data; PS, TP, JP, II, KML-G, RAF, Acquisition of data, Analysis and interpretation of data, Contributed unpublished essential data or reagents; MRB, JMB, Conception and design, Drafting or revising the article

## Additional files

### Major datasets

The following previously published datasets were used:

| Author(s) | Year | Dataset title | Dataset ID and/or URL | Database, license, and accessibility information |
|---|---|---|---|---|
| Oyama T, Ishino S, Fujino S, Ogino H, Shirai T, Mayanagi K, Saito M, Nagasawa N, Ishino Y, Morikawa K | 2011 | A protein complex essential initiation of DNA replication | http://www.rcsb.org/pdb/explore/explore.do?structureId=3ANW | Publicly available at RCSB Protein Data Bank. |
| Fletcher RJ, Bishop BE, Leon RP, Sclafani RA, Ogata CM, Chen XS | 2003 | The dodecamer structure of mcm from archaeal M. thermoautotrophicum | http://www.rcsb.org/pdb/explore/explore.do?structureId=1LTL | Publicly available at RCSB Protein Data Bank. |
| Liu W, Pucci B, Rossi M, Pisani FM, Ladenstein R | 2008 | Structural analysis of the Sulfolobus solfataricus MCM protein N- terminal domain | http://www.rcsb.org/pdb/explore/explore.do?structureId=2VL6 | Publicly available at RCSB Protein Data Bank. |
| Brewster AS, Wang G, Yu X, Greenleaf WB, Carazo JM, Tjajadia M, Klein MG, Chen XS | 2008 | Crystal Structure Of A Near Full-Length Archaeal MCM: Functional Insights For An AAA + Hexameric Helicase | http://www.rcsb.org/pdb/explore/explore.do?structureId=3F9V | Publicly available at RCSB Protein Data Bank. |
| Yamagata A, Kakuta Y, Masui R, Fukuyama K | 2002 | Crystal structure of exonuclease RecJ bound to manganese | http://www.rcsb.org/pdb/explore/explore.do?structureId=1IR6 | Publicly available at RCSB Protein Data Bank. |

| Chang YP, Wang G, Bermudez V, Hurwitz J, Chen XS | 2007 | The crystal structure of full length human GINS complex | http://www.rcsb.org/pdb/explore/explore.do?structureId=2Q9Q | Publicly available at RCSB Protein Data Bank. |
|---|---|---|---|---|
| Itsathitphaisarn O, Wing RA, Eliason WK, Wang J, Steitz TA | 2012 | A New Twist on the Translocation Mechanism of Helicases from the Structure of DnaB with its Substrates | http://www.rcsb.org/pdb/explore/explore.do?structureId=4ESV | Publicly available at RCSB Protein Data Bank. |
| Enemark EJ, Joshua-Tor L | 2006 | Crystal structure of papillomavirus E1 hexameric helicase with ssDNA and MgADP | http://www.rcsb.org/pdb/explore/explore.do?structureId=2GXA | Publicly available at RCSB Protein Data Bank. |

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
