## [Decision Letter]

Thank you for sending your work entitled “DNA binding polarity, dimerization, and ATPase ring remodeling in the Cdc45-MCM-GINS replicative helicase” for consideration at *eLife*. Your article has been favorably evaluated by James Manley (Senior editor) and 2 reviewers, one of whom is a member of our Board of Reviewing Editors.

The Reviewing editor and the other reviewers discussed their comments before we reached this decision, and the Reviewing editor has assembled the following comments to help you prepare a revised submission.

Overall, the experiments are well executed the data are sound and the main conclusions are justified. However, neither the inference regarding the ssDNA path (on Cdc45) nor the final model (in terms of conformational changes during the loading and translocation) are fully supported by the data. These issues can be addressed provided that the authors can address the following major points; additional experimentation is likely not required:

1) Relationship of ssDNA binding to Mcm subunits. Figure 2 shows a very specific structure in which DNA is uniquely in contact with the Mcm5 subunit. The basis for this figure is EM reconstructions of the CMG complex with bound DNA that contains a streptavidin tag. As ssDNA is poorly visualized under EM, the streptavidin marks the location of the DNA in the complex. However, 1) the class averages in Figure 2 show the streptavidin at a considerable distance from the CMG complex, making assignment of a specific DNA-binding MCM subunit difficult, and 2) as nearly as this reviewer can tell, there is nothing in the text that specifically identifies Mcm5 as the sole DNA binding subunit. Explain: is Figure 2 a structural summary or artistic license?

2) Thresholding/filtering issues. A major point in the paper is the physical interaction between CDC45 and Mcm5, in which the N terminus of CDC45 is proposed to essentially pry apart the N and C-terminal domains of Mcm5 (Figure 1). There are several technical problems with this conclusion. Although varying the thresholding (presumably through Chimera) is a useful way to emphasize various features, in Figure 1 it is so extensive that the N and C-terminal domains of Mcm5 are completely separated from one another, a feature not demonstrated by other CMG structures presented in this paper. In general, the thresholding level should generate an enclosed volume consistent with the calculated molecular weight of the component proteins: was this done? When 2 structures are being compared, the levels of thresholding and filtering should be the same in each structure.

3) In addition, the interpretation that CDC45 pries apart the N and C-terminus of Mcm5 largely depends upon how the various masses within the structure were segmented – what part of the density belongs to Mcm5 and what part actually belongs to CDC45? This question has an additional problem insofar as the region of CDC45 that likely interacts with Mcm5 is not highly conserved and the authors were unable to generate a homology model for it. Moreover, the software used for segmentation that sets the subunit boundaries (likely the Segger tool in Chimera) requires a very high-resolution structure to give reliable results without computational validation (< 10 angstroms). Although for transmission EM the structures in this manuscript are excellent, it seems a little optimistic that the authors have enough resolution to accurately validate these boundaries visually. There are computational methods available to provide proper validation of their results (http://ncmi.bcm.edu/ncmi/software/segger/docs_fitting_scores).

4) The figures are confusing and should be improved. Specifically:

A) In all the figures, top and side views should be kept in similar orientations whenever possible for clarity.

B) Figure 3 and Figure 6 are inconsistent in terms of leading strand and lagging strand positions relative to Mcm subunits.

C) Figure 3 suggests a rocking motion (rotation) between the CMG and the duplex DNA. This is not supported by data here. It is quite possible that the leading strand could slip out to the outer channel formed by GINS and Cdc45 without altering the relative orientation between duplex DNA and the CMG. Furthermore, the conformation when the leading strand is captured by Cdc45 is different from that shown here. This figure should be altered to reflect this.

D) Figure 5 suggests the exact register of the CMG dimer based on 2D class averages. Although it is plausible, there are no experimental data to confirm this and Models 3 or 5 cannot be completely dismissed. Furthermore, Figure 5 suggest there might be some flexibility in the dimer arrangement as not all the class averages display the same tight packing. This part of the Results/discussions should be tuned down.

E) Figure 6. Translocation step. The interactions of GINS and Cdc45 with Mcm are different in the two complexes. Please check polarity! The earlier steps neither are unclear nor supported by data here so should be omitted. Furthermore, since Mcm2-7 likely act sequentially, different subunits will have different nucleotide bound states at a given time. ATP-gS state presented here therefore might not reflect the functional state during translocation. It is also unclear what triggers gate open/close. How does the Cdt1-Orc complex (which induces the gate being blocked) enable dsDNA loading? What is the defining factor that leads to DNA melting and gate opening/strand separation? These are not addressed here and therefore Figure 6 should be modified to only reflect major conclusions here. The leading strand and lacking strand as well as their directionality should be labelled.

5) Methods: how many particles contributed to the final reconstruction? How were the streptavidin-labelled data processed? Is a 3D reconstruction obtained (since 436 micrographs were collected)?

6) Significance: the author argued that the need for GINS/Cdc45 in preventing DNA from escaping Mcm2-7 would likely to be minimal. This raises the question of the significance of the additional roles proposed in this paper (to capture the “escaped” leading strand). Please reconcile.

---

## [Author Response]

*1) Relationship of ssDNA binding to Mcm subunits.*
Figure 2
*shows a very specific structure in which DNA is uniquely in contact with the Mcm5 subunit. The basis for this figure is EM reconstructions of the CMG complex with bound DNA that contains a streptavidin tag. As ssDNA is poorly visualized under EM, the streptavidin marks the location of the DNA in the complex. However, 1) the class averages in*
Figure 2
*show the streptavidin at a considerable distance from the CMG complex, making assignment of a specific DNA-binding MCM subunit difficult, and 2) as nearly as this reviewer can tell, there is nothing in the text that specifically identifies Mcm5 as the sole DNA binding subunit. Explain - is*
Figure 2
*a structural summary or artistic license?*

We thank the reviewers for identifying a passage that was unclear in the figure legends. What is shown in Figure 2 is indeed density from the 3DEM experiment. The caption to Figure 2 has been modified to clarify this point. The rod-like density on top of the AAA+ Mcm tier had never been seen before in apo or ATPγS-bound CMG particles and we assign it to DNA because: *i)* the size and shape of the density is compatible with a 20mer duplex DNA stretch, and *ii)* the overall orientation of the duplex DNA end is confirmed by streptavidin labeling (which further allows us to localize the duplex end toward the C-terminal face of the Mcm ring). Regarding whether Mcm5 is the sole DNA-contacting subunit at the fork nexus, we have removed mention of this possibility in the text to avoid overinterpreting our data.

*2) Thresholding/filtering issues. A major point in the paper is the physical interaction between CDC45 and Mcm5, in which the N terminus of CDC45 is proposed to essentially pry apart the N and C-terminal domains of Mcm5 (*Figure 1*). There are several technical problems with this conclusion. Although varying the thresholding (presumably through Chimera) is a useful way to emphasize various features, in*
Figure 1
*it is so extensive that the N and C-terminal domains of Mcm5 are completely separated from one another, a feature not demonstrated by other CMG structures presented in this paper. In general, the thresholding level should generate an enclosed volume consistent with the calculated molecular weight of the component proteins: was this done? When 2 structures are being compared, the levels of thresholding and filtering should be the same in each structure*.

We thank the reviewers for touching on an important point and apologize if we might have inadvertently created any confusion around this issue; however, it is the combined action of the GINS•Cdc45 complex that appears to wedge apart the Mcm5 N- and C-terminal regions. Insofar as thresholding, all of the renderings mentioned above have indeed been prepared to consistently account for the molecular weight of the full CMG complex, with the exception of Figure 3—figure supplement 1 (where a slightly higher threshold highlights the Cdc45^CTD^-MCM contact) and Figure 2 (where a lower threshold was used to visualize the region assigned to duplex DNA – negative staining of nucleic acids generally results in “thinner” densities than what one would get with cryo-electron microscopy, which is why the threshold was lowered in this instance). Using the constant threshold shown in the primary figures, the separation of Mcm5 N- and C-terminal domains is apparent in all CMG renderings. For example, the CMG side-view shown in Figure 4 highlights the N-terminal and C-terminal-AAA+ domains separation of Mcm5 as in Figure 1. A similar separation of the N- and C-terminal domains of Mcm5 is likewise present in Figure 3; this splaying is only unapparent because Cdc45 is rendered opaque, occluding the view of Mcm5.

*3) In addition, the interpretation that CDC45 pries apart the N and C-terminus of Mcm5 largely depends upon how the various masses within the structure were segmented – what part of the density belongs to Mcm5 and what part actually belongs to CDC45? This question has an additional problem insofar as the region of CDC45 that likely interacts with Mcm5 is not highly conserved and the authors were unable to generate a homology model for it. Moreover, the software used for segmentation that sets the subunit boundaries (likely the Segger tool in Chimera) requires a very high-resolution structure to give reliable results without computational validation (< 10 angstroms). Although for transmission EM the structures in this manuscript are excellent, it seems a little optimistic that the authors have enough resolution to accurately validate these boundaries visually. There are computational methods available to provide proper validation of their results (*http://ncmi.bcm.edu/ncmi/software/segger/docs_fitting_scores*)*.

We thank the referees for the advice on segmentation. We agree that we did not include sufficient data to explain the rationale for our assignment of the C-terminal domain of Cdc45. We have now included a comparison between the N-terminal view of the isolated *Drosophila melanogaster* Mcm2-7 (as published in Lyubimov, *et al.* PNAS 2012) and the bottom view of the DNA-bound CMG (now included as Figure 3—figure supplement 1). This comparison shows that no density is appended to either the N-terminal helical domain of Mcm5 or Mcm2 in the isolated Mcm2-7 complex. This result led us to conclude that the density seen to interdigitate between Mcm5 and Mcm2 likely corresponds to the otherwise unaccountable C-terminal domain of Cdc45. However, we acknowledge that it is possible that the extreme N-terminus of Mcm2 (which is thought to be unstructured) might instead become ordered to occupy this region upon Cdc45 binding. We have modified the main text to account for this possibility.

4) The figures are confusing and should be improved. Specifically:

*A) In all the figures, top and side views should be kept in similar orientations whenever possible for clarity*.

All pertinent views have been rotated.

*B)*
Figure 3
*and*
Figure 6
*are inconsistent in terms of leading strand and lagging strand positions relative to Mcm subunits*.

We thank the referees for catching this mistake and have modified Figure 6 accordingly.

*C)*
Figure 3
*suggests a rocking motion (rotation) between the CMG and the duplex DNA. This is not supported by data here. It is quite possible that the leading strand could slip out to the outer channel formed by GINS and Cdc45 without altering the relative orientation between duplex DNA and the CMG. Furthermore, the conformation when the leading strand is captured by Cdc45 is different from that shown here. This figure should be altered to reflect this*.

We agree – Figure 3 has been reworked to clarify these issues.

*D)*
Figure 5
*suggests the exact register of the CMG dimer based on 2D class averages. Although it is plausible, there are no experimental data to confirm this and Models 3 or 5 cannot be completely dismissed. Furthermore,*
Figure 5
*suggest there might be some flexibility in the dimer arrangement as not all the class averages display the same tight packing. This part of the Results/discussions should be tuned down*.

The text and caption to Figure 5 have been modified to present the more nuanced view suggested by the referees. The flexibility evident in the dimer is also now mentioned.

*E)*
Figure 6*. Translocation step. The interactions of GINS and Cdc45 with Mcm are different in the two complexes. Please check polarity! The earlier steps neither are unclear nor supported by data here so should be omitted. Furthermore, since Mcm2-7 likely act sequentially, different subunits will have different nucleotide bound states at a given time. ATP-gS state presented here therefore might not reflect the functional state during translocation. It is also unclear what triggers gate open/close. How does the Cdt1-Orc complex (which induces the gate being blocked) enable dsDNA loading? What is the defining factor that leads to DNA melting and gate opening/strand separation? These are not addressed here and therefore*
Figure 6
*should be modified to only reflect major conclusions here. The leading strand and lacking strand as well as their directionality should be labelled*.

We would like to thank the reviewers for helping us get this complex figure right. We have modified the model to focus on the data presented in this paper and in other paper that have directly commented on the ring status of Mcm2-7. The leading and lagging strands and their directionality, along with GINS•Cdc45 and Mcm2-7 interactions, have been adjusted and are now more clearly labeled.

5) Methods: how many particles contributed to the final reconstruction? How were the streptavidin-labelled data processed? Is a 3D reconstruction obtained (since 436 micrographs were collected)?

The Methods have been amended to address these questions. A 3D structure of the streptavidin-labeled complex was indeed generated (see below); however, owing the positional mobility of the streptavidin, the reconstruction is no more informative than the reference-free class averages. Accordingly, we show the analysis here for the reviewers, but have elected not to include it in the manuscript. We can add this figure as supplemental data if the referees feel it important to do so.Author response image 1.Analysis of the streptavidin labeled CMG-DNA structure. (a) 2D class averages of the streptavidin labeled CMG-DNA complex (biotin is on the duplex end). (b) Structure of the streptavidin labeled CMG at 5.2 σ allows for discrimination between N-terminal and C-terminal tiers of the Mcm2-7. (c) Three-dimensional structure of the streptavidin bound CMG-DNA displayed at 1.5 σ highlights the presence of additional density surmounting the C-terminal Mcm2-7 tier. This density is roughly compatible in shape and size with a streptavidin tetramer (shown in green), but is poorly ordered due to the positional mobility of the tag.

*6) Significance: the author argued that the need for GINS/Cdc45 in preventing DNA from escaping Mcm2-7 would likely to be minimal. This raises the question of the significance of the additional roles proposed in this paper (to capture the “escaped” leading strand). Please reconcile*.

We apologize for failing to frame this issue more clearly, which was likely compounded by the confusion surrounding Figure 3 (which has now been reworked). In stating that the need for Cdc45 in preventing DNA escape is minimal, we should have said that we expect the frequency with which Cdc45 would be called upon to capture an escaped leading strand to be low. The rationale for this statement is because once bound to both ATP and DNA for translocation, it seems likely that the Mcm2/5 gate would open relatively infrequently (except, perhaps during fork stalling); however, whenever the gate does open, Cdc45 would be present to prevent DNA escape. We have modified the text to better highlight this distinction.